# RNase 7 Inhibits Uropathogenic *Escherichia coli*-Induced Inflammation in Bladder Cells under a High-Glucose Environment by Regulating the JAK/STAT Signaling Pathway

**DOI:** 10.3390/ijms23095156

**Published:** 2022-05-05

**Authors:** Chen-Hsun Ho, Pin-Wen Liao, Chia-Kwung Fan, Shih-Ping Liu, Po-Ching Cheng

**Affiliations:** 1Division of Urology, Department of Surgery, Shin Kong Wu Ho-Su Memorial Hospital, Taipei 11101, Taiwan; m015695@ms.skh.org.tw; 2School of Medicine, Fu Jen Catholic University, New Taipei City 24205, Taiwan; d620109001@tmu.edu.tw; 3Department of Neurology, Cathay General Hospital, Taipei 10630, Taiwan; 4Graduate Institute of Neural Regenerative Medicine, College of Medicine Science and Technology, Taipei Medical University, Taipei 11031, Taiwan; 5Department of Molecular Parasitology and Tropical Diseases, School of Medicine, College of Medicine, Taipei Medical University, Taipei 11031, Taiwan; tedfan@tmu.edu.tw; 6Center for International Tropical Medicine, College of Medicine, Taipei Medical University, Taipei 11031, Taiwan; 7Department of Urology, National Taiwan University Hospital and College of Medicine, Taipei 10002, Taiwan

**Keywords:** urinary tract infection, antimicrobial peptide, RNase 7, uropathogenic *Escherichia coli*, JAK/STAT signaling pathway, inflammatory responses

## Abstract

Antimicrobial peptides (AMPs), which are natural antibiotics, protect against pathogens invading the urinary tract. RNase 7 with antimicrobial properties has rapid and powerful suppressive effects against Gram-positive and Gram-negative bacterial infections. However, its detailed antibacterial mechanisms have not been fully determined. Here, we investigate whether RNase 7 had an impact on bladder cells under uropathogenic *Escherichia coli* (UPEC) infection in a high-glucose environment using in vitro GFP-UPEC-infected bladder cell and PE-labeled TLR4, STAT1, and STAT3 models. We provide evidence of the suppressive effects of RNase 7 on UPEC infection and UPEC-induced inflammatory responses by regulating the JAK/STAT signaling pathway using JAK inhibitor and STAT inhibitor blocking experiments. Pretreatment with different concentrations of RNase 7 for 24 h concentration-dependently suppressed UPEC invasion in bladder cells (5 μg/mL reducing 45%; 25 μg/mL reducing 60%). The expressions of TLR4, STAT1, and STAT3 were also downregulated in a concentration-dependent manner after RNase 7 pretreatment (5 μg/mL reducing 35%, 54% and 35%; 25 μg/mL reducing 60%, 75% and 64%, respectively). RNase 7-induced decrease in UPEC infection in a high-glucose environment not only downregulated the expression of TLR4 protein and the JAK/STAT signaling pathway but also decreased UPEC-induced secretion of exogenous inflammatory IL-6 and IL-8 cytokines, although IL-8 levels increased in the 25 μg/mL RNase 7-treated group. Thus, inhibition of STAT affected pSTAT1, pSTAT3, and TLR4 expression, as well as proinflammatory IL-6 and IFN-γ expression. Notably, blocking JAK resulted in the rebound expression of related proteins, especially pSTAT1, TLR4, and IL-6. The present study showed the suppressive effects of RNase 7 on UPEC infection and induced inflammation in bladder epithelial cells in a high-glucose environment. RNase 7 may be an anti-inflammatory and anti-infective mediator in bladder cells by downregulating the JAK/STAT signaling pathway and may be beneficial in treating cystitis in DM patients. These results will help clarify the correlation between AMP production and UTI, identify the relationship between urinary tract infection and diabetes in UTI patients, and develop novel diagnostics or possible treatments targeting RNase 7.

## 1. Introduction

Urinary tract infections (UTIs) are among the most common infections. The prevalence of UTIs in women is three times that in men. According to statistics in Taiwan, 40–50% of women suffer from UTI at least once in their lifetime [1,2]. Other high-risk groups include children, the elderly, patients with abnormal urinary tract structures, and those who receive urethral interventions such as catheterization [3]. The occurrence and severity of UTI can be related to genetics, hormones, behavioral factors, and toxicity of invading microorganisms, mainly uropathogenic *Escherichia coli* (UPEC) [4]. UTIs (80–90%) are often caused by *E. coli* that appears in the rectum; in particular, UPEC can exist in the uroepithelial cells after invading the urinary epithelium, which can easily cause recurrent UTIs [5]. Pathogens in relapse are usually resistant to antibiotic treatments.

Recent studies have shown that antimicrobial peptides (AMPs), which are natural antibiotics, protect against pathogens that invade the urinary tract. AMPs are small proteins produced by white blood cells and epithelial cells, when the innate immune system is attacked by pathogens [4]. AMPs in the urinary tract include defensins, cathelicidin, hepcidin, and RNase 7, an enzyme with antimicrobial properties that has rapid and powerful antibacterial effects against Gram-positive and Gram-negative bacterial infections. RNase 7 and others in the kidney and upper urinary tract also include THP, lactoferrin, lipocalin, and secretory leukocyte proteinase inhibitor (SLPI) [4,6]. The antibacterial function of AMPs is related to their charge, secondary structure, and hydrophobicity. In addition, some AMPs can pass through the cell membrane and affect the synthesis of proteins or DNA [6,7]. Because of these effects, AMPs are considered to be potential treatments for drug-resistant bacteria. Since microorganisms do not easily change their cell membrane structure and can maintain sensitivity to AMPs, AMPs can exhibit antimicrobial function at low concentrations [8]. AMPs not only complement the shortcomings of antibiotics but also have synergistic effects with them [9]. RNase 7 was initially found in the epidermis and was subsequently detected in the bladder, ureter, and kidney [10,11]. The concentration of RNase 7 in the urinary tract is higher than that of other AMPs and shows a sufficient bactericidal effect. The rapid and powerful antibacterial effect of RNase 7 in Gram-positive and Gram-negative bacterial infections is caused by the breakdown of cell membranes [11]. However, its detailed antibacterial mechanisms have not been fully determined.

Immunocompromised individuals have a higher risk of developing UTIs, and the symptoms of infection can differ from those of the general population [12]. Diabetes mellitus (DM) is considered a risk factor for UTI, and the causative pathogens of DM are also slightly different from those in the general population [12,13]. Minoru et al. found that patients with higher HbA1c and lower albumin levels required long treatment for UTIs [13]. Clinical studies have shown that blood levels of AMPs, which play an important role in the innate immune system, are positively correlated with plasma inflammatory markers in patients with type 2 diabetes and negatively correlated with HDL cholesterol [14]. The downregulation of RNase 7 can make the skin of DM2 patients prone to infection and affect the healing of ulcer wounds [15]. Eichler et al. confirmed that when insulin is bound to IR receptors on the bladder urothelium and kidney cells, the intracellular PI3K/AKT pathway is activated and induces RNase 7 production. RNase 7 can quickly kill urinary tract bacteria, thereby inhibiting the growth of UPEC and protecting urothelial cells [16]. Since DM is related to insufficient insulin production and resistance, and insulin can regulate the production of AMP, insulin treatment not only affects the metabolism of blood glucose in diabetic patients but also regulates the production of many immune proteins [14,15,16]. Our previous study confirmed that the ability of UPEC to invade bladder epithelial cell infection and induce cellular inflammation was significantly enhanced in a high-glucose environment. These responses were also found to be regulated by a mechanism of upregulation of the TLR4-involved pathway and the synergistic expansion of the JAK/STAT1 signaling pathway. Studies have also confirmed that TLR4 expression and IL-1β secretion in bladder cells are significantly increased under high glucose conditions, and lead to enhanced inflammation [17]. Our another study also showed that UPEC infection of prostate cells led to the activation of the JAK/STAT1 signaling pathway, and using testosterone administration effectively inhibited the UPEC invasion of prostate cells and induced inflammatory responses [18]. The regulation of the JAK2/STAT3 pathway in the skeletal muscle of patients with type 2 DM is closely related to the occurrence of insulin resistance, and the phosphorylation of JAK2 and STAT3 can increase the incidence of glucose tolerance loss or DM. In addition, STAT3 induces TLR4 production and further regulates proinflammatory cytokines and insulin resistance in myoblasts [19,20]. Our latest study also showed that insulin downregulated the infection of bladder epithelial cells caused by UPEC in a high-glucose environment, and its regulation was also obviously regulated by the JAK/STAT1 signaling pathway [21].

In summary, the present study aimed to demonstrate whether RNase 7 had an impact on bladder cells under UPEC infection in a high-glucose environment by regulating the JAK/STAT signaling pathway. We hypothesized that pretreatment of bladder cells with RNase 7 reduced UPEC invasion and induced-inflammatory responses, and that this inhibition might act through downregulation of the JAK/STAT pathway. We also hope that this study will further clarify the mechanism by which RNase 7 affects UPEC infection of bladder cells and inhibits inflammation. This study hopes to determine the association between UTIs and RNase 7 and to develop new therapies targeting AMPs to prevent the occurrence of UTI in patients with DM.

## 2. Results

### 2.1. RNase 7 Suppressed the Glucose-Enhanced UPEC Infection in Bladder Epithelial Cells

To confirm that RNase 7 reduced UPEC infection in bladder epithelial cells in a high-glucose environment, SV-HUC-1 cells pretreated with 15 mM glucose and following different concentrations of RNase 7 treatment were co-cultured with UPEC, and then detected by using colony formation assay. After co-cultivation with UPECs, extracellular bacteria were washed away, cells were lysed, and cell lysates were spread on LB agars to count invading UPECs. The results showed that RNase 7 decreased the glucose-enhanced UPEC infection rate in bladder cells, and the suppression was significantly observed in the 5 and 25 μg/mL pretreated groups (*p* < 0.0001 compared to the 15 mM glucose-treated group) (Figure 1A,B). The colony assay also revealed that the infection rate of the pretreated 25 μg/mL RNase 7 group was even lower than that of the UPEC infection control group (*p* < 0.01) (Figure 1B).

### 2.2. Decrease in UPEC Infection in Bladder Epithelial Cells by RNase 7 Was Related to Downregulation of STAT1, STAT3, and TLR4 Expression

The expressions of STAT1 and STAT3 in UPEC-infected SVHUC-1 cells with RNase 7 pretreatment for 24 h were downregulated in a concentration-dependent manner in the 5 and 25 μg/mL pretreated groups (*p* < 0.001 compared to the 15 mM glucose-treated group) (Figure 2 and Figure 3). The expression of STAT1 and STAT3 in the pretreated 25 μg/mL RNase 7 groups was also lower than that in the UPEC infection control group. In addition, RNase 7 inhibition appeared to be more potent for STAT1, as STAT1 expression was significantly lower in the 5 μg/mL pretreated group than in the UPEC control group (*p* < 0.05). As shown in Figure 4, the ratio of TLR4-expressed bladder epithelial cells also decreased with the concentration of pretreated RNase 7. The level of suppression in the 5 and 25 μg/mL pretreated groups was significantly reduced compared to that in the 15 mM glucose-treated group (*p* < 0.01), and that in the 25 μg/mL RNase 7 pretreatment was even lower than that in the UPEC-infected control group (*p* < 0.001). The present results demonstrate that RNase 7 not only reduced UPEC infection but also downregulates the expression of STAT1, STAT3, and TLR4 in bladder cells in a high-glucose environment.

### 2.3. RNase 7 Decreased UPEC Infection-Induced Exogenous Inflammatory Cytokine Secretion

To further understand the role of RNase 7 pretreatment in interfering with UPEC infection and inducing an inflammatory response in bladder cells, we determined whether RNase 7 pretreatment led to a reduction in the exogenous secretion of UPEC-induced inflammatory cytokines in bladder cells. A CBA analysis of the supernatants from cell cultures exposed to different concentrations of RNase 7 confirmed the inhibitory effects of RNase 7 on UPEC infection-stimulated inflammation (Figure 5). In the 1 and 5 μg/mL RNase 7-pretreated groups, the UPEC-stimulated secretion of extracellular IL-8 and IL-6 significantly decreased as compared to that of the 15 mM glucose-treated group (*p* < 0.01, *p* < 0.0001). Notably, extracellular IL-8 level in the 25 μg/mL RNase 7-pretreated group markedly increased, whereas the IL-6 level in the 25 μg/mL RNase 7-pretreated group was still lower than that of the 15 mM glucose-treated group (*p* < 0.0001), although higher than that of the UPEC infection control group (*p* < 0.001). On the other hand, although IL-1β exhibited a downward trend after pretreatment with RNase 7, there was no significant difference compared to the 15 mM glucose-treated group. In addition, the secretion of IL-10 by infected bladder cells was completely independent of RNase 7 pretreatment, showing no statistical correlation between the two. The results showed that both IL-8 and IL-6 inflammatory cytokines secreted from UPEC-infected bladder cells under a high-glucose situation were significantly reduced after RNase 7 pretreatment (as comparing to G groups).

### 2.4. RNase 7 Suppressed the Enhanced Expression of TLR4 and JAK/STAT Signaling Pathway in Bladder Cells with UPEC Infection in a Concentration-Dependent Manner

As shown in Figure 6, RNase 7-induced downregulation did not reflect only in the expression of TLR4 protein and inflammatory IL-6 and IFN-γ in the UPEC infection but also in the regulation of JAK/STAT pathway-related proteins. Significant differences were observed in the groups pretreated with RNase 7 that concentration-dependently reduced the IFN-γ, IL-6, TLR4, JAK1/2, STAT1, STAT3, and phosphorylated STAT1/3 levels (*p* < 0.05, compared to the 15 mM glucose-treated groups). It also enhanced the JAK/STAT pathway inhibitor SOCS3, especially in 25 μg/mL RNase 7 pretreatment (*p* < 0.05, compared to the 15 mM glucose-treated groups). The results clearly provided evidence that RNase 7 pretreatment might effectively reduce the level of the proinflammatory IFN-γ and IL-6 cytokines in UPEC-infected bladder cells by suppressing the TLR4-mediated and JAK/STAT signaling pathways.

### 2.5. RNase 7 Decreased the UPEC Infection in Bladder Epithelial Cells through Regulation of the JAK/STAT1 Signaling Pathway

Our results showed that RNase 7 might decrease UPEC infection in bladder epithelial cells by regulating the JAK/STAT signaling pathway in a high-glucose environment. To further confirm that RNase 7 suppressed UPEC infection in bladder epithelial cells through this signaling pathway, SV-HUV-1 cells were co-incubated with either a JAK or STAT inhibitor (25 and 50 mM/mL, respectively) following RNase 7 pretreatment to block the signaling pathway and then infecting the cells with glucose and UPEC as described. Twenty-four hours post-infection, all infected cells were lysed and plated on LB agar and measured to analyze UPEC colonization. As shown in Figure 7, pretreatment of bladder cells with RNase 7 after 15 μg/mL glucose pretreatment for 24 h significantly decreased the ability of UPEC infection enhanced by glucose (*p* < 0.001). However, co-incubation with either JAK1 or STAT1 inhibitor following RNase 7 pretreatment only revealed a slight increase in UPEC infection without statistical significance. In addition, when compared to co-incubation with the inhibitor alone, the infection rate of either JAK1 or STAT1 inhibitor treatment was significantly higher without RNase 7 pretreatment (*p* < 0.01, compared to AMP plus JAK1 or STAT1 inhibitor treatment, respectively).

### 2.6. Increased UPEC Infection in Bladder Epithelial Cells in a High-Glucose Environment Was Inhibited by RNase 7 Downregulating the JAK/STAT Pathway-Related Proteins

Figure 8 shows the expression of IFN-γ, IL-6, TLR4, and JAK/STAT pathway-related proteins in SV-HUC-1 cells co-incubated with either JAK or STAT inhibitor under RNase 7 pretreatment with UPEC infection. The results revealed that there was significant difference in the expression of IFN-γ, TLR4, and phosphorylated STAT1/3, as well as STAT3 in the group with STAT inhibitor co-incubation, regardless of RNase 7 treatment compared to the 15 mM glucose pretreated group (*p* < 0.05); whereas IL-6, JAK1/2, and STAT1 levels had no statistical difference after STAT inhibitor co-incubation, even with slightly elevated fluctuations. However, co-incubation with the JAK inhibitor did not exhibit different results with or without RNase 7 pretreatmentin in TLR4, IL-6, STAT1, and pSTAT1 expressions as compared to the 15 mM glucose-treated group. The other hand, it showed a difference to the RNase 7-pretreated group in the expression of TLR4, IL-6, and pSTAT1/3 (*p* < 0.05) when co-incubating with the JAK inhibitor; and the same in the STAT1/3 when co-incubating with the STAT inhibitor (*p* < 0.05). In addition, SOCS3 in either JAK or STAT inhibitor co-incubated groups without RNase pretreatment showed similar but slightly decreased expression levels than in the RNase 7-pretreated alone group. These results proved that RNase 7 decreased UPEC infection in bladder epithelial cells in a high-glucose environment by downregulating JAK/STAT pathway-related proteins, especially JAK.

## 3. Discussion

A previous study has shown that insulin contributes to host defense by regulating RNase 7 production and that urinary RNase 7 concentrations are suppressed in patients with insulin-deficient, new-onset type 1 DM and increase with insulin therapy [16]. Our laboratory recently used human urothelial cell culture models to demonstrate that insulin was an important hormone that suppressed UPEC infection in urothelial cells via the JAK/STAT signaling pathway [21]. The interaction mechanism between these two effects is worth exploring. In this study, we provided evidence of the suppressive effects of RNase 7 on UPEC infection in bladder cells and UPEC-induced inflammatory responses by regulating the JAK/STAT signaling pathway.

Antimicrobial peptides are soluble cell-derived mediators of natural defense mechanisms for cleaning the adherence and invasion of UPEC in the uroepithelium [22,23]. The present study clearly indicated that RNase 7 could suppress UPEC infection in bladder cells in a concentration-dependent manner in a high-glucose environment (Figure 1), and the 25 μg/mL RNase 7-pretreated group significantly declined compared the general infected control. Our results also confirmed that the inhibitory effect of RNase 7 on UPEC infection in bladder cells was positively correlated with the expression of STAT1, STAT3, and TLR4 proteins in the cells, especially STAT1 (Figure 2, Figure 3 and Figure 4). This is consistent with our previous findings that UPEC infection in bladder cells in high-glucose environments induces increased expression of TLR4 and the JAK/STAT pathway [17]. TLR4 is known to play a role in bacterial invasion, with TLR4/cAMP-mediated immune functions causing UPEC to be expelled from infected uroepithelial cells [24]. We previously showed that STAT1 and STAT3 played critical roles in the regulation of UPEC infection and colonization in uroepithelial cells [18,25]. Therefore, RNase 7 may inhibit the UPEC infection of bladder cells by regulating the expression of these proteins.

Although the anti-inflammatory effects of RNase 7 on microbial infection in a variety of tissues are well-known [26,27], the mechanism of action of RNase 7 in UPEC-induced bladder inflammatory responses remains unclear. Our data revealed that RNase 7 in bladder cells under UPEC infection in a high-glucose environment significantly inhibited JAK/STAT pathway-related JAK1/2, STAT1/3, and active phosphorylated STAT1/3, as well as the expression of TLR4 and proinflammatory IFN-γ and IL-6. JAK/STAT pathway-inhibitory SOCS3 expression also increased in the 25 μg/mL RNase 7-treated group and was significantly different from that of the 15 mM glucose-treated group. Simanski et al. used a STAT3 inhibitor and siRNA-mediated downregulation of STAT3, resulting in diminished IL-17A/IFN-c-mediated RNase 7 production in keratinocytes [27]. A clinical study found that children with UTI had an increased prevalence of RNase 7 genetic variations SNP rs1263872, which might increase UTI susceptibility by suppressing RNase 7 antibacterial activity against UPEC. Wild-type RNase 7 overexpression in human urothelial cultures reduces invasive UPEC infection [28]. We showed that RNase 7 could effectively inhibit the production and secretion of proinflammatory cytokines IL-6, IL-1β, and IL-8 by downregulating JAK1/2 and STAT1/3 expression (Figure 5). However, whether increased RNase 7 pretreatment or diminished invasion of UPEC is the reason causing inflammatory factors to decrease remains to be shown. Further, there was an abnormal increase in secreted IL-8 in the 25 μg/mL RNase 7-treated group. EGFR activation promotes EGFR-dependent expression of proinflammatory innate immune mediators such as IL-8 and CCL20, facilitating epithelial wound healing in airway epithelial cells. In addition, epithelial injury is a potent inducer of RNase 7 expression, which is induced by direct EGFR activation [26]. Whether high concentrations of RNase 7 also affect EGFR expression and cell repair in the bladder epithelium, which in turn leads to innate immunity such as IL-8 secretion, requires further investigation.

Inhibition of JAK/STAT signaling by androgen involves invasion and persistence of UPEC within prostate cells [18,25]. Inhibition of the JAK/STAT signaling pathway downregulates bacteria-induced HMGB1 cytoplasmic accumulation to promote autophagy and suppress bacterial phagocytosis [29,30,31]. Our data revealed that inhibition of JAK or STAT did not affect RNase 7-induced suppression of UPEC infection in bladder cells compared to the JAK inhibitor or STAT inhibitor-treated alone group (Figure 7). Moreover, subsequent protein expression experiments revealed that inhibition of STAT affected STAT3, pSTAT1, pSTAT3, and TLR4 expression, as well as proinflammatory IL-6 and IFN-γ expression. Notably, blocking JAK resulted in the rebound expression of related proteins as RNase 7 treatment, especially pSTAT1/3, TLR4, and IL-6 (Figure 8). This result is also consistent with that of our previous study [18]. The present study suggests that blocking JAK/STAT signaling transduction cannot reduce UPEC infection in bladder cells in a high-glucose environment, whereas the suppressive effects of RNase 7 on UPEC to infect bladder cells are not completely mediated by the JAK/STAT pathway. However, JAK regulation seems to be more associated with the suppression of RNase 7 to UPEC infection in bladder cells and requires further study.

## 4. Materials and Methods

### 4.1. Cell Culture

SV-HUC-1 cell line—a normal ureter epithelial cell line—was acquired from the American Type Tissue Culture Collection (ATCC, Rockville, MA, USA). The cells were maintained in Ham’s F-12K (Kaighn’s medium) supplemented with 10% (*v*/*v*) fetal bovine serum (Gibco; Invitrogen, Grand Island, NY, USA) at 37 °C in a humidified atmosphere of 5% CO_2_. The medium was replaced every 2 or 3 days.

### 4.2. Bacterial Strain

UPEC CFT073 from ATCC 700928 with fluorescent pGFP was used as the model organism [18]. Bacterial growth was determined spectrophotometrically at an optical density of 600 nm (OD600). For in vitro infection, the bacteria were suspended in culture medium at multiplicity of infection (MOI) of 100.

### 4.3. RNase 7 Treatments

In UPEC infection model, SV-HUC-1 cells were seeded into 24-well dishes (4 × 105 cells/well), and on the subsequent day were pretreated with 15 mM glucose for 24 h and then treated with RNase 7 (MyBioSource, San Diego, CA, USA) at different concentrations (1, 5, and 25 ug/mL) for 24 h, then replaced in new media prior to bacterial infection. Each experiment was performed at least thrice. Signal-blocking experiments were executed by pretreatment with 25 μM JAK inhibitor (Ruxolitinib; MedChem Express, Princeton, NJ, USA) or 50 μM STAT inhibitor (Fludarabine; Selleck Chemicals Inc., Houston, TX, USA) with glucose for 24 h before UPEC infection. The effects of RNase 7 treatment alone to UPEC and SV-HUC-1 cells are determined in Appendix A, respectively.

### 4.4. UPEC Infections and Quantifying 

SV-HUC-1 cells after pretreatment were replaced in new media and then incubated with UPEC bacteria (MOI of 100) for 4 h at 37 °C with 5% CO_2_. After the 4 h incubation period, the infected monolayers were washed 4 times with phosphate-buffered saline (PBS) and incubated for 30 min in growth medium containing gentamicin (100 μg/mL; Sigma-Aldrich, St. Louis, MO, USA). To measure bacterial invasion, the cells were lysed and harvested using 0.5% trypsin (Gibco)–0.1% Triton X-100 (Amresco, Solon, OH, USA), and then plated onto LB agar. Colonies were counted to quantify bound bacteria after 24 h of incubation. To detect the green fluorescence of UPEC, SV-HUC-1 cells seeded onto 18 mm coverslip (inside 12 well dishes) were infected with pGFP-UPEC (MOI of 100) and observed by fluorescence microscopy as described previously [32]. Human anti-STAT1, STAT3 and TLR4 phycoerythrin (PE,)-conjugated antibody (BD Biosciences, catalog number: 558537, 560391, 564215) were used. At least three coverslips per condition were examined. To quantify invasion, images of 20 random fields of each coverslip were acquired and counted with the help of computer image analysis software (ImageJ software) (version 1.41o) (National Institutes of Health, Bethesda, MD, USA) (http://rsb.info.nih.gov/ij/, accessed on 2 October 2012). Related proteins staining of untreated cells were determined in Appendix A as the referred data.

### 4.5. Cytometric Bead Array Immunoassay 

Medium of cell cultures were collected and centrifuged (13,000× *g*, 20 min, 4 °C), and the supernatant was assessed using a human inflammatory cytokines cytometric bead array (CBA, BD Biosciences, San Diego, CA, USA) for cytokines IL-8, IL-10, IL-1β, and IL-6. The cytokine capture bead, phycoerythrin (PE) detection reagent and recombinant standards or test samples were incubated together for 3 h at room temperature. We used a FACSCanto flow cytometer (BD Biosciences, San Diego, CA, USA) to acquire the data and analyzed by BD CBA Analysis Software to receive the graph [33,34].

### 4.6. Cell Protein Extraction

The treated cells were centrifuged at 4 °C, 1800 rpm for 5 min, and the supernatant was poured out and placed in RIPA lysis buffer (containing 0.5 M Tris-HCl, pH 7.4, 1.5 M NaCl, 2.5% deoxycholic acid, 10% NP-40, 10 mM EDTA) (Pierce, Rockford, IL, USA). The cells were homogenized with a microgrinder and then mixed evenly to collect the expressed protein. After all the samples were collected, cell extracts were centrifuged at 12,000× *g* rpm at 4 °C for 10 min, and the supernatant was collected to obtain total soluble proteins. All protein extracts were stored in the refrigerator at −80 °C for later use.

### 4.7. Western Blotting

The proteins in the sample were harvested as described above. The experiment procedure was also following our previous publication [18]. The antibodies used were listed as below: anti-JAK1, anti-JAK2, anti-STAT1, anti-pSTAT1, anti-STAT3, anti-pSTAT3 (Cell Signaling, Beverly, MA, USA; catalog number:#3344, #3230, #9172, #7649, #9139, #9136); anti-SOCS3 (GeneTex, Irvine, CA, USA; catalog number:GTX23693); anti-IFN-γ (Abcam, Cambridge, UK; catalog number:ab9657); anti-TLR-4 (Proteintech, Rosemont, IL, USA; catalog number:19811-1-AP); anti-IL-6 (Bioworld Technology, Bloomington, MN, USA; catalog number:BS6419); at room temperature (RT) for 1 h. The protein bands on the membrane were detected with an ECL-Plus Western Blot Detection system (GE Healthcare UK LTD) according to the instructions of the manufacturer. All experiments were replicated at least thrice.

### 4.8. Statistical Analysis

The study was exploratory, and descriptive statistical analysis was executed in the research. The data were expressed as mean ± standard deviation (SD). Analysis of variance (ANOVA) was used to analyze the significant differences of all the data collected. A post hoc test between pairwise groups was determined using the Bonferroni corrected *t*-test. The differences between various-concentration RNase 7 treatment groups and positive controls (15 mM glucose pretreatment) were mainly compared. The significance threshold was set at 0.05 and was considered as a statistical difference.

## 5. Conclusions

The present study showed the suppressive effects of RNase 7 on UPEC infection and induced inflammation in bladder epithelial cells in a high-glucose environment. Our findings indicate that Rnase 7 may be an anti-inflammatory and antibacterial mediator in bladder cells by downregulating the JAK/STAT signaling pathway and may be beneficial in treating cystitis in DM patients.

## Figures and Tables

**Figure 1 ijms-23-05156-f001:**
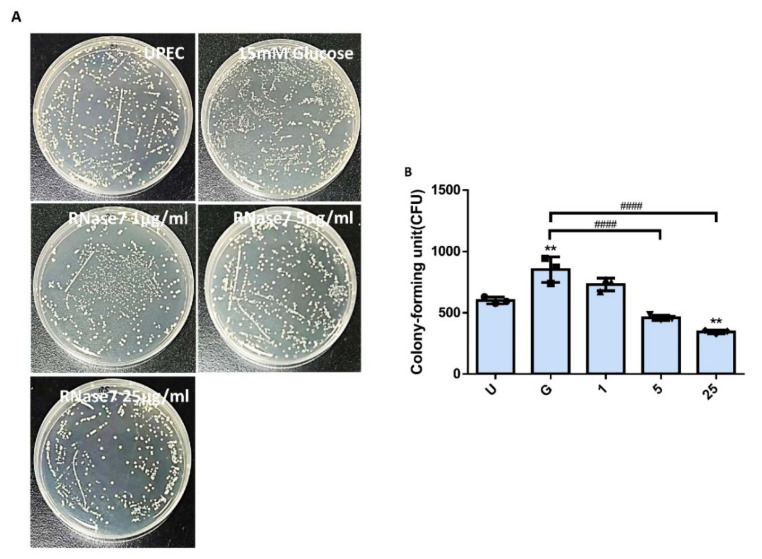
Influence of RNase 7 on UPEC infection in bladder cells in a high-glucose environment. SV-HUC-1 cells were pretreated with different concentrations of RNase 7 (1, 5, and 25 μg/mL) and 15 mM glucose for 24 h and then infected with pGFP-UPEC (MOI:1:100) as described in the text. UPEC infection in bladder cells was examined by plating on LB agar (**A**) and determined using ImageJ software (**B**). Colony-forming units (CFUs) were acquired after plating out lysed solutions of infected cells. The definitions of each letter/number represented by the *X* axis are: U: UPEC infection alone; G: with 15 mM glucose pretreatment alone; 1, 5, and 25: pretreated with 1, 5, and 25 μg/mL RNase 7, respectively, + 15 mM glucose. Data are expressed as the mean ± SD of three independent experiments. ** *p* < 0.01, compared to UPEC infection control groups. #### *p* < 0.0001, compared between the two groups.

**Figure 2 ijms-23-05156-f002:**
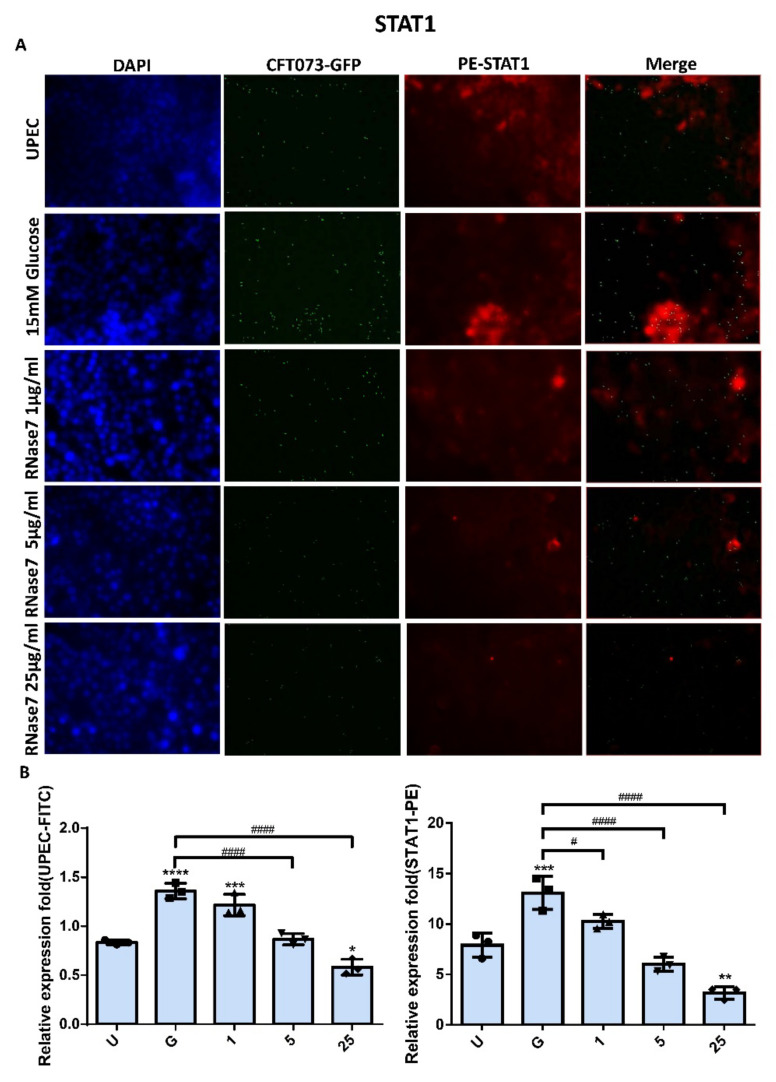
RNase 7 affected UPEC infection and STAT1 expression in bladder cells. After 24 h of pretreatment with different concentrations of RNase 7 (1, 5, and 25 μg/mL) and 15 mM glucose, GFP-UPEC and PE-STAT1 expression in SV-HUC-1 cells post-infection were (**A**) observed by fluorescence microscopy and (**B**) measured using ImageJ software. Cells infected with UPEC alone were used as the positive controls. The data shown are representative of a typical result. DAPI was used to count the number of cells and as a standard for calculating the fluorescence expression ratio of cells. Cell images were captured using a microscope (Leica) at 100× magnification. The definitions of each letter/number represented by the *X* axis are: U: UPEC infection alone; G: with 15 mM glucose pretreatment alone; 1, 5, and 25: pretreated with 1, 5, and 25 μg/mL RNase 7, respectively, + 15 mM glucose. Data are expressed as the mean ± SD of three independent experiments. * *p* < 0.05, ** *p* < 0.01, *** *p* < 0.001, **** *p* < 0.0001, compared to UPEC infection control groups. # *p* < 0.05, #### *p* < 0.0001, compared between the two groups.

**Figure 3 ijms-23-05156-f003:**
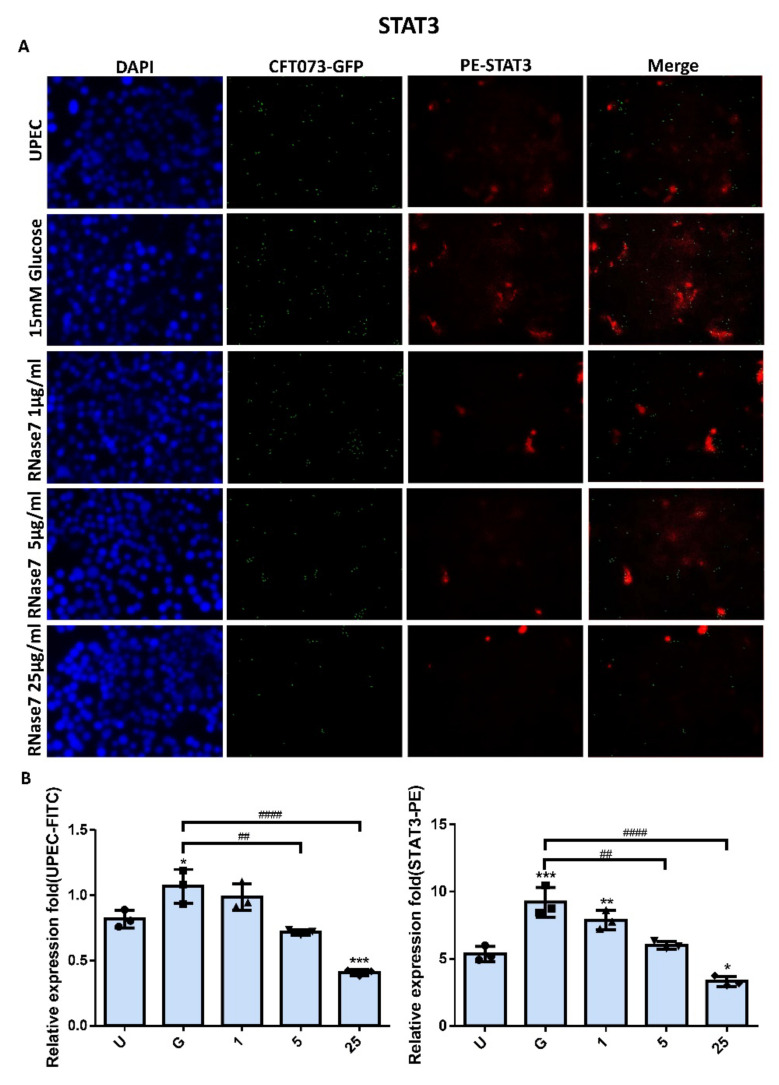
RNase 7 affected UPEC infection and STAT3 expression in bladder cells. After 24 h of pretreatment with different concentrations of RNase 7 (1, 5, and 25 μg/mL) and 15 mM glucose, GFP-UPEC and PE-STAT3 expression in SV-HUC-1 cells post-infection were (**A**) observed by fluorescence microscopy and (**B**) measured using ImageJ software. Cells infected with UPEC alone were used as the positive controls. The data shown are representative of a typical result. DAPI was used to count the number of cells and as a standard for calculating the fluorescence expression ratio of cells. Cell images were captured using a microscope (Leica) at 100× magnification. The definitions of each letter/number represented by the *X* axis are: U: UPEC infection alone; G: with 15 mM glucose pretreatment alone; 1, 5, and 25: pretreated with 1, 5, and 25 μg/mL RNase 7, respectively, + 15 mM glucose. Data are expressed as the mean ± SD of three independent experiments. * *p* < 0.05, ** *p* < 0.01, *** *p* < 0.001, compared to UPEC infection control groups. ## *p* < 0.01, #### *p* < 0.0001, compared between the two groups.

**Figure 4 ijms-23-05156-f004:**
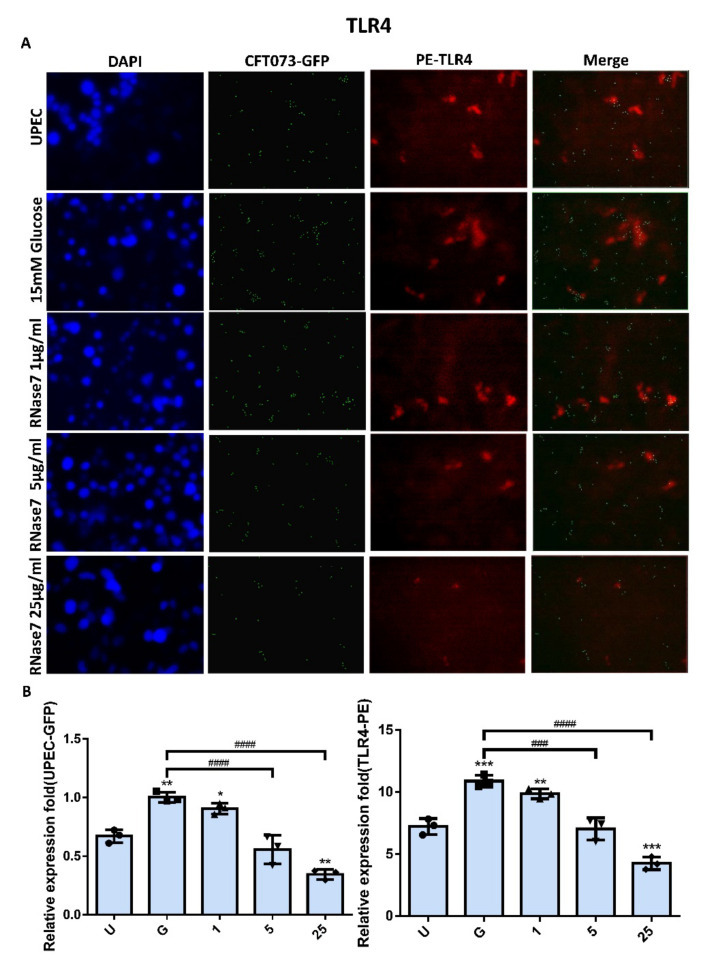
RNase 7 affected UPEC infection and TLR4 expression in bladder cells. After 24 h of pretreatment with different concentrations of RNase 7 (1, 5, and 25 μg/mL) and 15 mM glucose, GFP-UPEC and PE-TLR4 expression in SV-HUC-1 cells post-infection were (**A**) observed by fluorescence microscopy and (**B**) measured using ImageJ software. Cells infected with UPEC alone were used as the positive controls. The data shown are representative of a typical result. DAPI was used to count the number of cells and as a standard for calculating the fluorescence expression ratio of cells. Cell images were captured using a microscope (Leica) at 100× magnification. The definitions of each letter/number represented by the *X* axis are: U: UPEC infection alone; G: with 15 mM glucose pretreatment alone; 1, 5, and 25: pretreated with 1, 5, and 25 μg/mL RNase 7, respectively, + 15 mM glucose. Data are expressed as the mean ± SD of three independent experiments. * *p* < 0.05, ** *p* < 0.01, *** *p* < 0.001, compared to UPEC infection control groups. ### *p* < 0.001, #### *p* < 0.0001, compared between the two groups.

**Figure 5 ijms-23-05156-f005:**
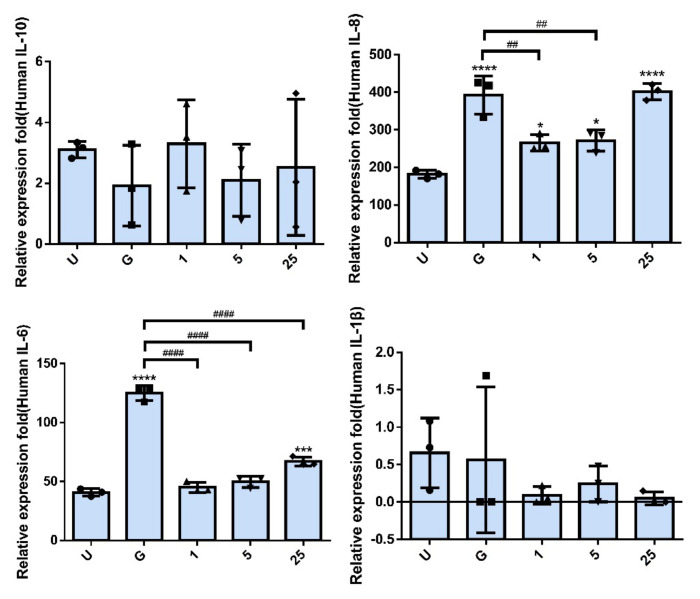
RNase 7 affected the secretion of inflammatory cytokines in UPEC-infected bladder cells. After treatment with different concentrations of RNase 7 (1, 5, and 25 μg/mL) and 15 mM glucose for 24 h, the release levels of cytokines IL−8, IL−10, IL−1β, and IL−6 from UPEC-infected SV-HUC-1 cells were measured using CBA. Cells infected with UPEC alone (MOI:1:100) were used as positive controls. The definitions of each letter/number represented by the *X* axis are: U: UPEC infection alone; G: with 15 mM glucose pretreatment alone; 1, 5, and 25: pretreated with 1, 5, and 25 μg/mL RNase 7, respectively, + 15 mM glucose. Data are expressed as the mean ± SD of three independent experiments. * *p* < 0.05, *** *p* < 0.001, **** *p* < 0.0001, compared to UPEC infection control groups. ## *p* < 0.01, #### *p* < 0.0001, compared between the two groups.

**Figure 6 ijms-23-05156-f006:**
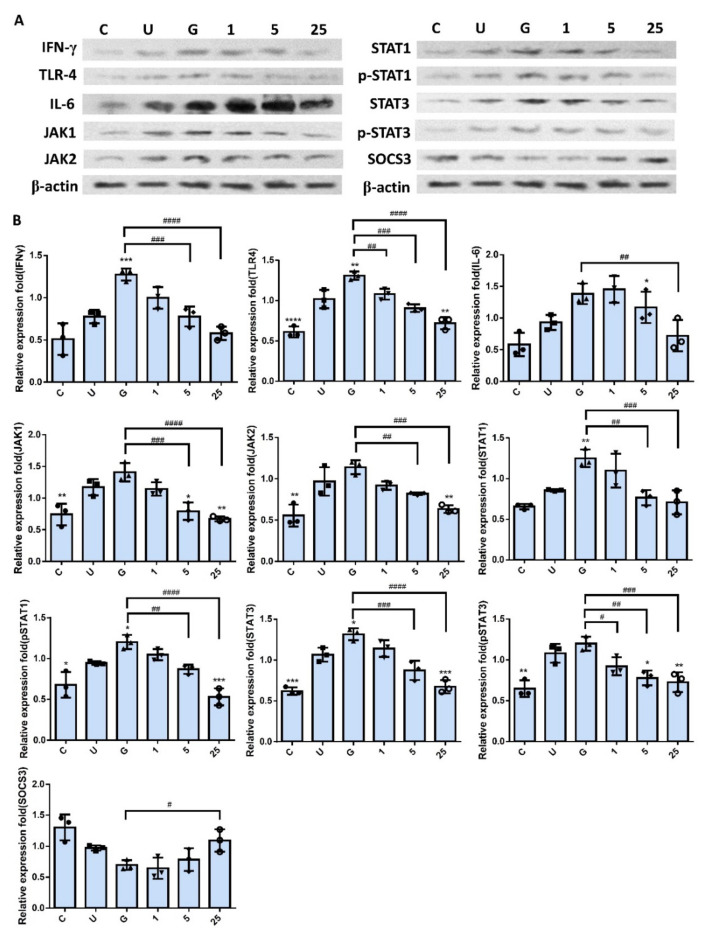
RNase 7 downregulated the expression of proteins associated with the JAK/STAT signaling pathway and inflammation in UPEC-infected bladder cells in a high-glucose environment. SV-HUC-1 cells were pretreated with different concentrations of RNase 7 (1, 5, and 25 μg/mL) and 15 mM glucose for 24 h and then infected with UPEC. Total protein from all the cell groups was collected for detection. (**A**) Total protein expression of JAK1, JAK2, STAT1, STAT3, phosphorylated-STAT1/STAT3, and the inhibitor, SOCS3, TLR4, IL-6, and IFN-γ were analyzed using Western blotting. (**B**) All data were normalized to the internal reference β-actin. Results were assessed using a densitometer and quantified using ImageJ software (NIH). The definitions of each letter/number represented by the *X* axis are: C: control; U: UPEC infection alone; G: with 15 mM glucose pretreatment alone; 1, 5, and 25: pretreated with 1, 5, and 25 μg/mL RNase 7, respectively, + 15 mM glucose. The results are presented as mean ± SD of three independent experiments. * *p* < 0.05, ** *p* < 0.01, *** *p* < 0.001, **** *p* < 0.0001, compared to the respective positive control group (U). # *p* < 0.05, ## *p* < 0.01, ### *p* < 0.001, #### *p* < 0.0001, compared between the two groups.

**Figure 7 ijms-23-05156-f007:**
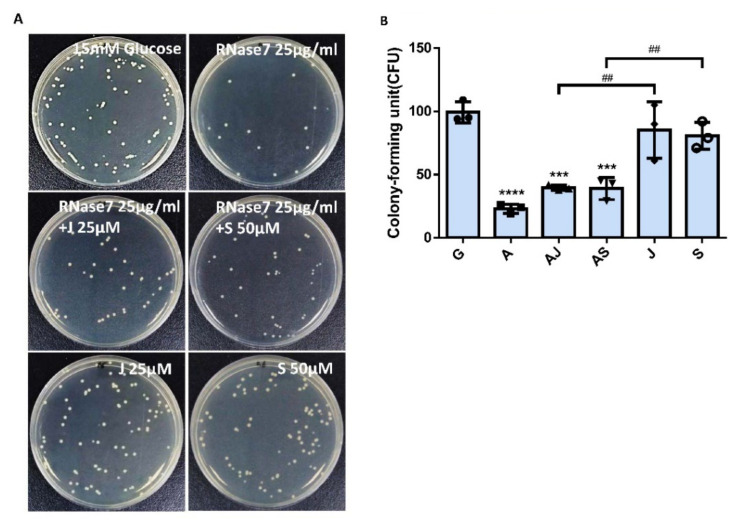
JAK and STAT inhibitors affected the suppression of RNase 7 in UPEC-infected bladder cells in a high-glucose environment. SV-HUC-1 cells were pretreated with 25 μg/mL RNase 7 and simultaneously with 25 μM JAK inhibitor or 50 μM STAT inhibitor for 24 h, followed by UPEC infection (MOI:100) as described above. Cells pretreated with 25 μM JAK inhibitor or STAT3 inhibitor alone for 24 h followed by UPEC infection were used as respective controls. Cells infected with 15 mM glucose and UPEC alone were used as the respective positive controls. The definitions of each letter represented by the *X* axis are: C: control; G: with 15 mM glucose pretreatment alone; A: pretreated with 25 μg/mL RNase 7; AJ: coincubated with 25 μg/mL RNase 7 and 25 μM JAK inhibitor; AS: coincubated with 25 μg/mL RNase 7 and 50 μM STAT inhibitor; J: cells pretreated with 25 μM JAK inhibitor alone; S: Cells pretreated with STAT inhibitor alone. The picture shown is representative of a typical result. Twenty-four hours post-infection, all infected cells were (**A**) lysed and plated on LB agar and (**B**) measured using ImageJ software to analyze UPEC colonization. Colony-forming units (CFUs) were acquired after plating 10-fold dilutions of infected cells with lysis. Data are expressed as mean ± SD of three separate experiments. *** *p* < 0.001, **** *p* < 0.0001, compared to UPEC infection control groups. ## *p* < 0.01, compared between the two indicated groups.

**Figure 8 ijms-23-05156-f008:**
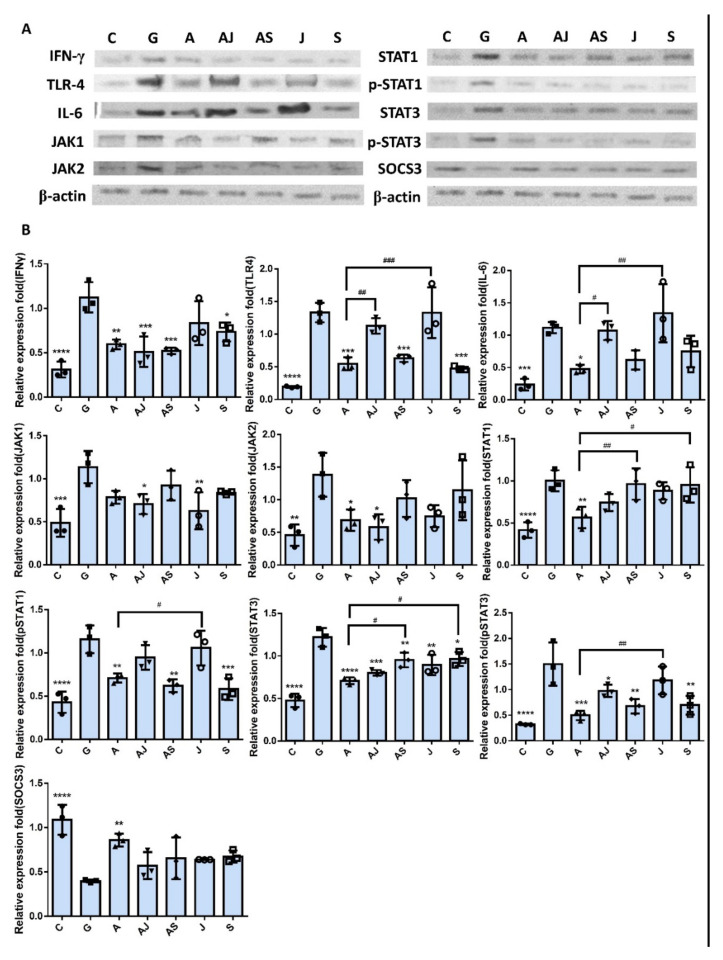
JAK and STAT inhibitors affected the RNase 7-downregulated protein expressions associated with the JAK/STAT signaling pathway and inflammation in UPEC-infected bladder cells in a high-glucose environment. SV-HUC-1 cells were pretreated with 15 mM glucose and coincubated with/without 25 μg/mL RNase 7, 25 μM JAK inhibitor, or 50 μM STAT inhibitor for 24 h, respectively. Then, all the cells were infected with UPEC (MOI:100) for following experiments. The definitions of each letter represented by the *X* axis are: C: control; G: with 15 mM glucose pretreatment alone; A: pretreated with 25 μg/mL RNase 7; AJ: coincubated with 25 μg/mL RNase 7 and 25 μM JAK inhibitor; AS: coincubated with 25 μg/mL RNase 7 and 50 μM STAT inhibitor; J: cells pretreated with 25 μM JAK inhibitor alone; S: Cells pretreated with STAT inhibitor alone. The picture shown is representative of a typical result. Total protein from all the cell groups was collected for detection. (**A**) Total protein expression of JAK1, JAK2, STAT1, STAT3, phosphorylated-STAT1/STAT3, and the inhibitor, SOCS3, TLR4, IL-6, and IFN-γ were analyzed using Western blotting. (**B**) All data were normalized to the internal reference β-actin. Results were assessed using a densitometer and quantified using ImageJ software (NIH). The results are presented as mean ± SD of three independent experiments. * *p* < 0.05, ** *p* < 0.01, *** *p* < 0.001, **** *p* < 0.0001, compared to the 15 mM glucose pretreated infection group. # *p* < 0.05, ## *p* < 0.01, ### *p* < 0.001, compared between the two groups.

## Data Availability

Not applicable.

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
