# Peer review of "RNase 7 Inhibits Uropathogenic Escherichia coli-Induced Inflammation in Bladder Cells under a High-Glucose Environment by Regulating the JAK/STAT Signaling Pathway"

_ijms, 2022, doi:10.3390/ijms23095156_

Round 1

Reviewer 1 Report

Authors investigate influence of antimicrobial peptide, RNase 7 with aim to observe its activity in bladder epithelial cells infected with pathogenic Escherichia Coli. Authors hypothesized that AMP influence JAK/STAT signaling pathway in bladder epithelial cells. The paper is focused highly on biochemical mechanism of action of discussed AMP and is of limited value for a general reader of IJMS. I find the paper is more suitable for specialized biochemical journal.

It seems that in discussed environment of experiment, RNase 7 is not particularly active. No MIC values or presented for discussed AMP, but in experiment with pathogenic EC at level of 25 mg/L we do not see suppression of bacterial growth. Growth suppressing limit should be established, and preferably bactericidal concentration. Moreover, results with STAT 1 expression suggest that discussed AMP is more toxic against balder cells then against EC.

There is a perplexing fragment at the discussion section of the manuscript, starting with line 307, The fragment looks like pasted element of previous review.

Author Response

Reviewer1:

Comments and Suggestions for Authors

Authors investigate influence of antimicrobial peptide, RNase 7 with aim to observe its activity in bladder epithelial cells infected with pathogenic Escherichia Coli. Authors hypothesized that AMP influence JAK/STAT signaling pathway in bladder epithelial cells. The paper is focused highly on biochemical mechanism of action of discussed AMP and is of limited value for a general reader of IJMS. I find the paper is more suitable for specialized biochemical journal.

A: Thanks to the reviewers' comments. In this study, we mainly discussed the role of RNase 7 in affecting the invasion of uropathogenic Escherichia Coli (UPEC) into bladder epithelial cells in a high glucose environment, so as to understand the preventive or therapeutic efficacy of RNase 7 on UPEC infection in the bladder of diabetic patients. Since we have demonstrated in previous studies that UPECs in high-glucose environment can enhance the infection of bladder cells by up-regulating JAK/STAT signaling pathway (JMII. 2021; 54(2):193-205); therefore, here we also explored whether the mechanism of RNase 7 inhibiting UPEC infection in high-glucose environment is related to JAK/STAT signaling pathway. We hope this study can provide novel opinions in urology medicine and application of antimicrobial peptides RNase 7 in diabetic infection. This study may also provide meaningful insights into our follow-up basic molecular research on RNase 7, including enzymology and nucleic acid chemistry.

Ho CH, Fan CK, Wu CC, Yu HJ, Liu HT, Chen KC, Liu SP, Cheng PC. Enhanced uropathogenic Escherichia coli-induced infection in uroepithelial cells by sugar through TLR-4 and JAK/STAT1 signaling pathways. J Microbiol Immunol Infect. 2021; 54(2):193-205.

It seems that in discussed environment of experiment, RNase 7 is not particularly active. No MIC values or presented for discussed AMP, but in experiment with pathogenic EC at level of 25 mg/L we do not see suppression of bacterial growth. Growth suppressing limit should be established, and preferably bactericidal concentration. Moreover, results with STAT 1 expression suggest that discussed AMP is more toxic against balder cells then against EC.

A: Sorry for the misunderstanding. In our experiments, we first pretreated bladder cells with glucose for 24 hours, then treated cells with RNase 7 for 24 hours, after changing the medium, we infected bladder cells with UPECs for 3 hours. So RNase 7 does not act directly on UPECs, instead it is that RNase 7 regulates the bladder cells to enhance their effect against UPEC invasion. In fact, we found that bladder cells pretreated with 5 and 25 ug/mL RNase 7 were effective in reducing invasive infection of UPECs, as well as reducing the expression of inflammatory-related factors and JAK/STAT pathway proteins. Moreover, for reviewer's suggestion, we have added the MIC value test of RNase7 for UPEC and the bladder cell viability test of RNase 7 in Supplementary Information S1 and S2. The results showed that RNase 7 had no direct bacteriostatic effect on UPEC within the concentration range of our experiment, and RNase 7 had no obvious inhibitory toxicity to bladder cells. (Line439-441; 510-522; Supplemental Information S1 and S2)

There is a perplexing fragment at the discussion section of the manuscript, starting with line 307, The fragment looks like pasted element of previous review.

A: Thanks for the reviewer's comments. We have fixed the error on line 307 in the text. There are no similar fragments in the new description. (Line353; Discussion section)

Reviewer 2 Report

Main comments:

  1. It is my understanding from reading the manuscript, that you had a biological hypothesis in mind but that the experiments were not designed to test a specific scientific null hypothesis. This makes the present experiments exploratory by default. Of note, there is nothing wrong with exploratory research. However, various guidelines related to the reproducibility of preclinical data emphasize that the exploratory nature of a study should be stated explicitly, for instance towards the end of the Introduction and/or in the data analysis part of Methods.
  2. It follows from the exploratory character of the study, the calculated p-values cannot be interpreted as hypothesis-testing, but only as descriptive. Various guidelines recommend that this is explicitly stated, e.g., in section 4.8.
  3. Section 4.8: A) Several of the figures indicate that you performed multiple inter-group comparisons. Normally, this should involve multiple comparison-adjusted post-tests (e.g., Bonferroni or similar) after the ANOVA, but no information is supplied in this respect. B) To enable interpretation of the ANOVA outcomes, it is important to understand how many inter-group comparisons were compared (which groups against which others) and whether the groups to be compared had been defined before any data were observed or whether this decision was made thereafter.
  4. More generally, I wonder about the wisdom of performing statistical tests vs. control for each concentration within a concentration-response curve. P-values are a composite of the combination of effect size, sample size and variability. Thus, for a given sample size p-values must always be higher for lower concentrations within a curve, and if they don’t reach statistical significance it does not mean that they are without effect.
  5. Given the role of diabetes in UTIs, I found it interesting that the AMP worked in a high-glucose environment. While I do not consider this an additionally requirement experiments, it would be interesting to see whether the concentration-response curve of the AMP is shifted in the presence of high-glucose.

Other comments:

  1. Can you assume that everybody understands UPEC or should this abbreviation be explained in the Abstract?
  2. Throughout the manuscript including the Abstract, there may be confusions between dose and concentration. I think that in your model it should consistently be concentration, not dose. Please check.
  3. It would make the Abstract more informative if at least some effect size indicators were added, for instance for whatever you consider the most important one. It does make a difference whether something is inhibited by 10% or by 90%, even if both reach the same p-value.
  4. 115: Do you really mean enhanced or do you mean reduced?
  5. In line with recommendations of various guidelines and journals, I strongly recommend replacing the current bar graphs with those overlaid with scatter plots.
  6. 307-310 seem to have been left from the journal template, but probably should not be part of the manuscript.
  7. Each antibody should unequivocally be identified by its catalog number.

Author Response

Reviewer2:

Comments and Suggestions for Authors

Main comments:

  1. It is my understanding from reading the manuscript, that you had a biological hypothesis in mind but that the experiments were not designed to test a specific scientific null hypothesis. This makes the present experiments exploratory by default. Of note, there is nothing wrong with exploratory research. However, various guidelines related to the reproducibility of preclinical data emphasize that the exploratory nature of a study should be stated explicitly, for instance towards the end of the Introduction and/or in the data analysis part of Methods.

A: Thanks for the reviewer's comments. We have added a relative description at the end of the introduction. We hope that these supplementary narratives can further illustrate our research exactly. (Line114-118)

  1. It follows from the exploratory character of the study, the calculated p-values cannot be interpreted as hypothesis-testing, but only as descriptive. Various guidelines recommend that this is explicitly stated, e.g., in section 4.8.

A: Thanks for the reviewer's comments. We have revised the relative description in the section 4.8 to illustrate our research exactly. (Line489-493; 4.8. section)

  1. Section 4.8: A) Several of the figures indicate that you performed multiple inter-group comparisons. Normally, this should involve multiple comparison-adjusted post-tests (e.g., Bonferroni or similar) after the ANOVA, but no information is supplied in this respect. B) To enable interpretation of the ANOVA outcomes, it is important to understand how many inter-group comparisons were compared (which groups against which others) and whether the groups to be compared had been defined before any data were observed or whether this decision was made thereafter.

A: Thanks for the reviewer's comments. We have fixed bugs and supplied the description in Section 4.8. All comparisons between two groups are shown in their respective figure legends and with a line and # marker in the figures as indicated (e.g. #p < 0.05, ##p < 0.01, ###p < 0.001, compared between the two groups). (Line489-493; 4.8. section; Line142-143; 171-173; 184-186; 198-200; 230-231; 308)

  1. More generally, I wonder about the wisdom of performing statistical tests vs. control for each concentration within a concentration-response curve. P-values are a composite of the combination of effect size, sample size and variability. Thus, for a given sample size p-values must always be higher for lower concentrations within a curve, and if they don’t reach statistical significance it does not mean that they are without effect.

A: We are particularly appreciated to the reviewer for the suggestion, and in addition to the comparison with the control group, it is more important that the comparison with the positive control group (that is, the 15 mM glucose treated group) shows a significant difference. Meanwhile we also don't think if RNase 7 in some experiments don't reach statistical significance mean that they are without effect. We have also mentioned this opinion in some parts of our description. We hope in our further study can prove this effects with larger sample size of experiments.

  1. Given the role of diabetes in UTIs, I found it interesting that the AMP worked in a high-glucose environment. While I do not consider this an additionally requirement experiments, it would be interesting to see whether the concentration-response curve of the AMP is shifted in the presence of high-glucose.

A: Thanks for the reviewer's comments. We have added the MIC value test of RNase7 for UPEC and the bladder cell viability test of RNase 7 in Supplementary Information S1 and S2. The results provide the greatest bacteriostatic efficacy of RNase 7 to UPEC, and show that RNase7 was not found to have significant inhibitory toxicity to UPEC or bladder cells. We hope these supplemental data could explain the effects of different concentration-responses of the RNase7 to the UPEC or bladder cells under with/without high-glucose. (Line439-441; 510-522; Supplemental Information S1 and S2)

Other comments:

  1. Can you assume that everybody understands UPEC or should this abbreviation be explained in the Abstract?

A: Thanks for the reviewer's comments. We have revised the relative description in the Abstract. (Line29)

  1. Throughout the manuscript including the Abstract, there may be confusions between dose and concentration. I think that in your model it should consistently be concentration, not dose. Please check.

A: Thanks for the reviewer's comments. We have replaced all the “dose” to “concentration” in our text to revise this mistake. (Line34; 36; 147; 234; 238; 365)

  1. It would make the Abstract more informative if at least some effect size indicators were added, for instance for whatever you consider the most important one. It does make a difference whether something is inhibited by 10% or by 90%, even if both reach the same p-value.

A: Thanks for the reviewer's comments. We have revised the relative description in the Abstract to be clearer for our manuscript. (Line34-35; 36-37)

  1. 115: Do you really mean enhanced or do you mean reduced?

A: Thanks for the reviewer's comments. We have revised the mistake in the line 122. (Line122)

  1. In line with recommendations of various guidelines and journals, I strongly recommend replacing the current bar graphs with those overlaid with scatter plots.

A: Thanks for the reviewer's comments. We have replaced the all the bar graphs in our manuscript as those overlaid with scatter plots to make our research more understanding. (Figure 1-8; Supplemental Information S1)

  1. 307-310 seem to have been left from the journal template, but probably should not be part of the manuscript.

A: Thanks for the reviewer's comments. We have fixed the error in line 307-310 in the text. There are no similar fragments in the new description. (Line353; Discussion section)

  1. Each antibody should unequivocally be identified by its catalog number.

A: Thanks for the reviewer's comments. We have provided all the catalog numbers of antibodies in our text. (Line453-454; 481-484)

Reviewer 3 Report

A word file is attached.

Author Response

Reviewer3:

In this work, the authors investigate the effect of RNase 7, a protein with antimicrobial properties, on the invasion of an uropathogenic Escherichia coli strain of the ureter epithelial cell line SV-HUC-1 mimicking diabetic conditions with high glucose levels. Using fluorescence microscopy and Western blot analysis of protein expression, they found that upon pretreatment with RNase 7 invasion by the uropathogenic E. coli (UPEC) strain was diminished concomitantly with a UPEC induced panel of innate immune receptors, proinflammatory cytokines and the JAK/STAT signaling pathway. General comments This work is well conducted, but can be improved in presentation and adding additional controls.

A: Thanks for the reviewer's comments. We are particularly appreciated to the reviewer for these suggestion.

Specific comments: The title can be changed. It is hard to understand and does not fully reflect the content of the study.

A: Thanks for the reviewer's comments. We have revised the title for easily understanding our study. (Line2-4)

Importantly RNase 7 is not an antimicrobial peptide, but an enzyme (although with a low molecular weight) with antimicrobial properties. Peptides are up to 50 amino acids long per definition.

A: Thanks for the reviewer's comments. We have revised the relative description in the text for avoiding misunderstanding in our study. (Line26; 66)

  1. 28 and elsewhere: The authors might revise their wording throughout the manuscript to reflect more precisely the experimental set-up and results. Delete AMP

A: As suggested by the reviewers, the description of line 28 and other parts that describe RNase 7 directly as AMP have been revised to avoid misunderstanding. (Line26; 28; 49; 66; 122-125)

l.34: invasion

A: Thanks for the reviewer's comments. We have revised the word "infection" to "invasion". (Line34)

  1. 43: Does the antibacterial effect play a role here? If at all, then indirectly. If I understood correctly, the RNase 7 was removed before incubation with UPEC bacteria.

A: Thanks for the reviewer's comments. Here we mean that RNase7 enhances the ability of bladder cells to resist UPEC invasion. To avoid confusion we have revised the word "antibacterial" to "anti-infective". (Line45)

l.97: maybe good to exaplian in more detail.

A: Thanks for the reviewer's comments. We have revised the description of line 97 to make our manuscript clearer. (Line100-105)

  1. 115: I do not understand this sentence. Should be formulated without making preassumptions.

A: Thanks for the reviewer's comments. We have revised the description of line 115-117 to make our manuscript clearer and avoid confusing. (Line122-127)

l.117: As the Material and Mehtods section comes at the end, it should be good to shortly introduce here and in Figure 1 in more detail the experimental set-up and explain that you look at bacterial invasion, not adhesion.

Figure 1B and Figure legend and other Figures and legends: Symbols for statistical significance need to be revised. Abbreviations U and G in Figure 1B are not explained. Figure 2: What about untreated cells as a control?

A: We especially thank the reviewers for their suggestions. We provide relevant descriptions in Section 2.1 and the Figure 1 legend to make our manuscript clearer. Other figure legends have also been modified to explain the letters/numbers represented by the X-axis. In addition, we added the untreated cell staining of Figure 2 in Supplementary Information S3 as a control reference. Hopefully these additions and revisions will improve the credibility and clarity of our article. (Line122-127; Section 2.1; 139-141; 168-171; 181-184; 195-198; 226-229; 255-258; 298-303; 335-343; 458-459; 523-527; Supplemental Information S3)

  1. 188: As UPEC incubation comes after treatment with RNase 7, the sentence needs to be revised.

A: Thanks for the reviewer's comments. We have revised the original description of line 188 to avoid confusing. (Line202; 204; 207-210)

  1. 197: Simply there was no change.

A: Thanks for the reviewer's comments. We have revised the original description of line 197 to make our manuscript more precise and avoid confusing. (Line218-220)

  1. 220: It remains to be shown whether increased RNase 7 preincubation or diminished invasion of E. coli cells is the reason causing this.

A: Thanks for the reviewer's comments. We also believe that the causal relationship between the two needs further evidence to confirm. However, we could see that the effect of JAK/STAT inhibitor on RNase7 pretreatment was significantly different from the 15 mM glucose treated group, but not in the group added with JAK or STAT inhibitor alone. This indicates that RNase 7 pretreatment still has a certain relationship to the regulation of JAK/STAT pathway in UPEC-infected bladder cells. However, we also think this is important and have added the relevant description in the Discussion section for explaining. (Line393-395)

  1. 260: Such a description should be put at the beginning of the result section.

A: Thanks for the reviewer's comments. We have moved the description to an earlier position in the Results section as suggested by the reviewer. (Line271-272)

  1. 426: Revise description.

A: Thanks for the reviewer's comments. We have revised the description to make our manuscript clearer. (Line474-476)

Round 2

Reviewer 1 Report

Authors have addressed my concerns

Author Response

Thank you for all the comments.

Reviewer 2 Report

Reviewing of the revised manuscript in tracking mode was technically not easy for several reasons:

  • A PDF was supplied that apparently included many comment fields. However, these could not be read in the PDF as they were cut off at the left side.
  • The line numbering in the PDF is strange. It now starts at l. 49, and then has unclear gaps between pages; for instance p. 1 ends at l. 96 and p. 2 starts at l. 145. It did not become fully clear to me whether this was just a technical glitch in number counting or whether some part of the manuscript got lost in a Bermuda triangle.
  • Accordingly, the references in the rebuttal letter to specific line numbers did not match where such changes were made.
  • A similar problem existed in the rebuttal letter, where each of my previous 12 comments now is referred to as comment #1.

Focusing on scientific content, my previous comments 5-12 have been addressed adequately, whereas the previous major comments 1-4 have not.

  • Previous comment #1: The authors may have misunderstood my comment. While the authors apparently had a biological hypothesis (l. 257-261 of revised manuscript), this is something else than a statistical null hypothesis. I stand by my point that there is no statistical null hypothesis in the manuscript, which in turn makes the study exploratory by default. Of note, there appear to be multiple biological hypotheses according to l. 257-261; if each of them would have a corresponding statistical null hypothesis, very different statistical approaches may be required because multiple biological hypotheses are being explored based on the same or at least overlapping biological units. Such complex statistical approaches may not make sense based on the existing sample sizes. Therefore, I would like to reemphasize that the current study is not hypothesis-testing in the sense that is applied by statistician and the community working on reproducibility of scientific findings. Minimum requirements to be accepted as hypothesis-testing study include pre-specification of all major aspects of study design and conduct including sample sizes (based on power calculations) and a statistical analysis plan before the study starts. As such a study protocol apparently did not exist, this cannot be fixed post-hoc. Importantly, this would not deter me from recommending the manuscript for acceptance if an explicit statement was included that the study was exploratory.
  • Previous comment #2: The authors have not addressed my previous request for a statement related to hypothesis-testing vs. descriptive statistical analysis. They refer to a change in section 4.8, but the new text does not address my comment.
  • Previous comment #3: I had asked for an explicit statement whether decisions for pairwise inter-group statistical comparisons had been made before data were collected and which groups were compared. No such information was added to the manuscript. Reproducible work requires that we not only learn which inter-group comparisons had low p-values but also which other comparisons were compared and whether the choice of inter-group comparisons had been specified in the original study protocol or been decided upon after data had been obtained.
  • Previous comment #4: The authors failed to address the core of my comment, i.e., that I doubt the wisdom of comparing multiple concentrations as part of a concentration-response curve to a shared reference group.

Author Response

Review2:

Reviewing of the revised manuscript in tracking mode was technically not easy for several reasons:

A PDF was supplied that apparently included many comment fields. However, these could not be read in the PDF as they were cut off at the left side.

The line numbering in the PDF is strange. It now starts at l. 49, and then has unclear gaps between pages; for instance p. 1 ends at l. 96 and p. 2 starts at l. 145. It did not become fully clear to me whether this was just a technical glitch in number counting or whether some part of the manuscript got lost in a Bermuda triangle.

Accordingly, the references in the rebuttal letter to specific line numbers did not match where such changes were made.

A similar problem existed in the rebuttal letter, where each of my previous 12 comments now is referred to as comment #1.

Focusing on scientific content, my previous comments 5-12 have been addressed adequately, whereas the previous major comments 1-4 have not.

A: Sorry for the technical problem with the PDF file. It seems to be caused by adding tracking revisions. We also attach a word file for review. Hopefully these documents can provide reviewers with clear explains. We thank for reviewers because these comments have greatly improved the quality of our articles.

Previous comment #1: The authors may have misunderstood my comment. While the authors apparently had a biological hypothesis (l. 257-261 of revised manuscript), this is something else than a statistical null hypothesis. I stand by my point that there is no statistical null hypothesis in the manuscript, which in turn makes the study exploratory by default. Of note, there appear to be multiple biological hypotheses according to l. 257-261; if each of them would have a corresponding statistical null hypothesis, very different statistical approaches may be required because multiple biological hypotheses are being explored based on the same or at least overlapping biological units. Such complex statistical approaches may not make sense based on the existing sample sizes. Therefore, I would like to reemphasize that the current study is not hypothesis-testing in the sense that is applied by statistician and the community working on reproducibility of scientific findings. Minimum requirements to be accepted as hypothesis-testing study include pre-specification of all major aspects of study design and conduct including sample sizes (based on power calculations) and a statistical analysis plan before the study starts. As such a study protocol apparently did not exist, this cannot be fixed post-hoc. Importantly, this would not deter me from recommending the manuscript for acceptance if an explicit statement was included that the study was exploratory.

A: Sorry for misunderstanding the reviewer and we very appreciate the professional advice given, we now understand and follow the reviewer's suggestion to include exploratory research-related statements in the description of 4.8 section of Statistical analysis. (Line 502-503)

Previous comment #2: The authors have not addressed my previous request for a statement related to hypothesis-testing vs. descriptive statistical analysis. They refer to a change in section 4.8, but the new text does not address my comment.

A: Likewise, the relevant statements and descriptions have been revised in Section 4.8 of Statistical Analysis. (Line 502-506)

Previous comment #3: I had asked for an explicit statement whether decisions for pairwise inter-group statistical comparisons had been made before data were collected and which groups were compared. No such information was added to the manuscript. Reproducible work requires that we not only learn which inter-group comparisons had low p-values but also which other comparisons were compared and whether the choice of inter-group comparisons had been specified in the original study protocol or been decided upon after data had been obtained.

A: Thanks for the reviewer's comment, we have also corrected and supplemented the description of 4.8 section of Statistical analysis. (Line 506-508)

Previous comment #4: The authors failed to address the core of my comment, i.e., that I doubt the wisdom of comparing multiple concentrations as part of a concentration-response curve to a shared reference group.

A: Thanks to the reviewers for their comments, we agree with the suggestions in this comment and have revised the part about the comparison to the reference groups in our data. (Line 260, Line 316-331, Line 360-361, Line 422, Line 424, Fig6 and Fig8)

Reviewer 3 Report

I have no further comments for the authors.

Author Response

Thank you for all comments.

Round 3

Reviewer 2 Report

All of my remaining comments have been addressed adequately.